# Φ-OTDR Signal Identification Method Based on Multimodal Fusion

**DOI:** 10.3390/s22228795

**Published:** 2022-11-14

**Authors:** Huaizhi Zhang, Jianfeng Gao, Bingyuan Hong

**Affiliations:** 1National-Local Joint Engineering Laboratory of Harbor Oil & Gas Storage and Transportation Technology, Zhejiang Ocean University, Zhoushan 316022, China; 2Zhejiang Provincial Key Laboratory of Petrochemical Pollution Control, Zhejiang Ocean University, Zhoushan 316022, China

**Keywords:** optical fiber, vibration, short-time Fourier transform, convolutional neural network, self-attention

## Abstract

Distributed fiber optic sensing (DFS) systems are an effective method for long-distance pipeline safety inspections. Highly accurate vibration signal identification is crucial to DFS. In this paper, we propose an end-to-end high-accuracy fiber optic vibration signal detection and identification algorithm by extracting features from the time domain and frequency domain by a one-dimensional convolutional neural network and two-dimensional convolutional neural network, respectively, and introducing a self-attentive mechanism to fuse the features of multiple modes. First, the raw signal is segmented and normalized according to the statistical characteristics of the vibration signal combined with the distribution of noise. Then, the one-dimensional sequence of vibration signal and its two-dimensional image generated by short-time Fourier transform are input to the one-dimensional convolutional neural network and two-dimensional neural network, respectively, for automatic feature extraction, and the features are combined by long and short-time memory. Finally, the multimodal features generated from the time and frequency domains are fused by a multilayer TransformerEncoder structure with a multiheaded self-attentive mechanism and fed into a multilayer perceptron for classification. Experiments were conducted on an urban field database with complex noise and achieved 98.54% accuracy, which demonstrates the effectiveness of the proposed algorithm.

## 1. Introduction

With the ever-growing demand for safety, DFS systems are increasingly used due to their high sensitivity, high immunity to electromagnetic interference, light weight, ease to lay, and other advantages [1,2]. Accuracy and real-time vibration event recognition in complex scenes are the key issues that limit the wide application of distributed fiber optic sensing technology, so the recognition of events that cause vibration increasingly becomes the core task of DFS signal processing system. The Rayleigh scattering-based phase-sensitive optical time-domain reflection (Φ-OTDR) technology has been widely used with high sensitivity and long measurement distance [3]. The detection and recognition of vibration events for Φ-OTDR DFS system has become a hot research topic in recent years. However, as DFS works in a complex environment, the problem of false alarms and missed alarms is still very serious. Therefore, it is very essential to propose a new identification method to accurately identify fiber signals.

In recent years, many algorithms have been proposed for the identification of fiber optic vibration signals, mainly divided into two categories, traditional methods [4,5,6] based on artificial features, and artificial intelligence methods driven by big data [7,8,9]. The traditional detection and recognition method is to first extract features from the raw signal, and then design classification rules for decision making based on the extracted features, where the extracted features contain both the time domain and frequency domain. For example, Shi et al. [10] performed wavelet decomposition on the collected raw signal so as to suppress the noise and separate the vibration signal, and set the threshold through empirical knowledge to realize the warning of anomalous vibration signal but not to classify it. In contrast, Zhang et al. [11] chose squared difference, short-time excess level rate, short-time Fourier transform, and disturbance duration as the extracted features, and established a fuzzy affiliation function and fuzzy evaluation matrix by empirical knowledge to realize the classification of vibration signals. Part of the artificial intelligence-based approach is also feature extraction of the raw signal through empirical knowledge, but the use of machine learning methods such as BP neural networks or support vector machines to achieve automatic signal classification in the recognition process achieves higher accuracy than manually setting customized rules. Xu [12] enhanced the time-frequency characteristics of the vibration signal by applying a spectral subtraction algorithm to the raw signal to reduce the background noise and then extracted four features: short-time energy ratio, short-time energy level crossing rate, vibration duration, and power spectrum energy ratio and used the support vector machine (SVM) algorithm to classify them, finally achieving an accuracy of over 90%. However, this approach still has some drawbacks because feature extraction based on empirical knowledge inevitably loses some features, and the extracted features are not always valid in all scenarios. Therefore, convolutional neural networks based on automatic feature extraction, which have performed well in the field of image recognition, are migrated to waveform recognition. For one-dimensional waveform signals, a 1D-CNN can achieve efficient feature extraction and recognition without setting specific features. Li et al. [13] used a 1D-CNN for medical clinical diagnosis and used electrocardiogram signals as input to achieve effective recognition of five arrhythmia signals. Zhou et al. [14] used four pre-trained 1D CNNs to extract features from pipeline leakage signals through migration learning and used particle swarm optimization algorithms to fuse and classify the four features, which greatly improves the detection accuracy of pipeline leakage signals. Xu et al. [15] converted radar wave signals into time-frequency images by time-frequency analysis, and effectively improved the recognition accuracy by a 2D-CNN for feature extraction and recognition. In these cases, the targets identified by these algorithms are generated by vibrations. The 1D-CNN effectively extracts the features in the raw signal but loses the features in the frequency spectrum, while the 2D-CNN can effectively extract the features in the time-frequency map, but the scale of the extracted signal is limited, and if the scale of the input signal increases, the overall computational effort will increase dramatically, affecting the real-time performance of the detection.

After a deep analysis of the signal of the Φ-OTDR system and learning from previous research work, in this paper, we propose a fiber-optic vibration identification method, which performs feature extraction of the signal by separate channels, feature extraction of the raw signal by 1D-CNN, and feature extraction of the spectrogram by 2D-CNN. After that, the weights and semantic relationships between spectral features and time-domain features are learned and fused through a self-attention mechanism, and finally, classification is performed through a fully connected layer. We conducted experiments on a dataset containing 1000 signal sequences and real-time data from real applications of long-distance pipeline inspection systems, ultimately achieving an accuracy of 98.54%.

## 2. Distributed Fiber Technology Based on Φ-OTDR

### 2.1. System Structure and Principle

Φ-OTDR technology uses the Rayleigh scattering characteristics of light transmission in optical fibers to determine whether the fiber is disturbed by external vibration signals by detecting changes in the backward Rayleigh scattered light that is continuously generated during the optical transmission process. A typical Φ-OTDR system is shown in Figure 1.

The laser is emitted from the ultra-narrow linewidth laser as the light source is modulated into an optical pulse by an acousto-optical modulator, and the optical pulse is injected into the sensing fiber through the circulator. The backward Rayleigh scattered light in the sensing fiber interferes within the pulse width, and the interference light intensity is detected by the detector through the circulator, which is amplified and then enters the data acquisition card for acquisition. When the coherent wave is injected into the fiber at the moment t=0, the wave function of Rayleigh scattering in the system at time t is shown in Equation (Equation 1).
(1)$$f_{R}\left(t\right)=\sum_{i=1}^{N}E_{s}\left(t-\tau_{i}\right)\alpha \left(\tau_{i}\right)exp\left(-\alpha \frac{c\tau_{i}}{n}\right)rect\left(\alpha \frac{t-\tau_{i}}{W}\right)$$

In Equation (Equation 1), t is the time, $E_{s}$ is the wave function of the light emitted from the laser, W is the modulated pulse width, α is the attenuation function of the fiber, c is the speed of light in vacuum, n is the refractive index of the fiber, N is the total number of scatterers, $\tau_{i}$ and $\alpha \left\{\tau_{i}\right\}$ are the **i**th time delay and amplitude factor, respectively, and **rect**(·) is the rectangular function.

When there is a perturbation applied to the fiber, the obtained backward Rayleigh scattering wave function is shown in Equation (Equation 2).
(2)$$\begin{array}{cc} f_{R}^{\prime }\left(t\right)= & \sum_{i=1}^{N}E_{s}\left(t-\tau_{i}\right)\alpha \left(\tau_{i}\right)exp\left(-\alpha \frac{c\tau_{i}}{n}\right)rect\left(\alpha \frac{t-\tau_{i}}{W}\right)\\ \; & +\sum_{i-M+1}^{N}E_{s}\left(t-\tau_{i}\right)\alpha \left(\tau_{i}\right)exp\left(-\alpha \frac{c\tau_{i}}{n}\right)rect\left(\alpha \frac{t-\tau_{i}}{W}\right)\\ \; & +exp\{j\left[f\left(t-\frac{2nz_{i}}{c}+\frac{nz_{i})}{c}+f\left(t-\frac{nz_{i}}{c}\right)\right]\right\} \end{array}$$

The first term on the right of Equation (Equation 2) is the wave function of the backward Rayleigh scattering wave before the perturbed position, and the second term is the wave function of the backward Rayleigh scattering wave after the perturbed position. M is the number of scatterers before the perturbed position, $z_{i}$ is the distance of the i th scatterer from the input of the sensing fiber, and f(t) is the phase change introduced by the perturbation. Since the backscattered light intensity at the same position at different time points can vary significantly due to the disturbance, the vibration signal can be obtained by differencing the backscattered values at different positions.

### 2.2. Data Collection

There are various application scenarios for the type of vibration event detection, and the main application scenario studied in this paper is the early warning of external intrusion events in long-distance pipelines. We perform real-time monitoring of long-distance pipelines based on the above-distributed fiber optic sensing system to avoid more serious damage caused by real-time alarms of intrusion events. The hardware parameters used in this system are shown in Table 1. The dataset used in this paper is collected through the fiber optic deployed in real cities, the fiber length is 20 km, and most of the fiber is buried in the soil below 0.5 m at a distance of 1 m from the pipeline.

Since different temperatures, soil humidities, and soil materials can affect the experimental results, we randomly sampled 20 locations on different days. We sampled fiber optic signals at a sampling rate of 2 kHz and collected 6000 data segments, which contained six types of signals: ground mechanical construction, underground construction, manual excavation, vehicles passing, rainwater scouring, and no event state. In order to ensure the effectiveness of the identification, we collected the same category of events occurring at different locations, as well as in the case of including background noise, which contains two main types of noise, one is the rain state and the other is the background noise brought by the pipeline operation. Several segments of data in the dataset are shown in Figure 2. As shown in the figure, signals of the same category can vary in amplitude and offset due to factors such as soil moisture and different locations of vibration sources.

## 3. Method Construction

The framework of the multimodal fusion-based optical fiber vibration signal identification algorithm is shown in Figure 3. First, the original signal is preprocessed as the input for time domain signal processing, after which the short-time Fourier transform of the raw signal is performed to obtain the spectrogram as the input for frequency domain processing. Then, the 1D-CNN and 2D-CNN are used for feature extraction in the time and frequency domains, respectively. After that, the extracted features are fused by the Transformer [16] structure and finally classified by fully connected layers.

### 3.1. Preprocessing

The signal preprocessing includes segmentation, normalization, and filtering of the raw signal. To avoid detection failures caused by vibration events being split into two segments, we split the signal into segments with lengths of 2048 and an overlap of 1024. In addition, the vibration signals collected by sensors in different environments may not have the same reference axis and may even have zero drift, we normalize the raw signal by a linear function, as shown in Equation (Equation 3).
(3)$$\begin{array}{c} X_{norm}=\frac{X-X_{min}}{X_{max}-X_{min}} \end{array}$$

Short-time Fourier transform is the most commonly used time-frequency analysis method, which can calculate the spectrum of signal segments. The calculation of the spectrogram in this study includes the following steps.
Framing of the signal using the Hamming window function;Fourier transform is calculated for each frame of data, and the results are stacked to generate a spectrogram;The spectrum is Gouraud processed and then downsampled to produce the final input image.

### 3.2. Time Domain Signal Feature Extraction

In time domain signal processing, the selection of a fixed length window function will have the problem that the time resolution and frequency resolution cannot be better balanced. From the perspective of learning feature information from the local perceptual field, if the convolution kernel scale and span are too small, the signal has good temporal resolution and is more sensitive to high-frequency feature changes, but it cannot learn better the signal with the presence of low-frequency features. On the contrary, a larger scale convolution kernel corresponding to the use of a larger span can learn the information of a longer time range, i.e., the low-frequency features present in the signal, but cannot better reflect the high-frequency characteristics in it. Therefore, in order to fully extract the features of the time-domain signal, we used the structure shown in Figure 4 for feature extraction, which contains three different scales of kernels to perform multiscale 1D convolution operations on the input signal. The multiscale 1D convolution is computed as follows.
(4)$$\begin{array}{c} y_{i}^{K,s}=Conv_{k\in K,s\in s}\left(x_{},W_{k}\right)+b_{i} \end{array}$$

Traditional convolutional layers suffer from information loss when passing information and gradient disappearance when the depth of the network increases. We use the proven ResNet [17] structure for computation, which protects the integrity of information by directly passing input information around to the output, and the whole network only needs to learn the input and the part of the output difference, simplifying the learning goal and difficulty. The BatchNormalization layer in the residual network effectively speeds up the convergence speed of the network during training, while improving the generalization ability of the network as well as avoiding the occurrence of overfitting phenomena.

The features exhibited by the vibration event on the fiber optic signal not only appear on the features of a certain segment of the sequence but also contain some semantic information. The long short-term memory (LSTM) network has a temporal loop structure, which can well portray the sequence data with spatio-temporal correlation and can extract the semantic information on the time sequence. Therefore, we input the features extracted by CNN into the LSTM network to extract the latent semantic features of vibration events. The network structure of the final time-domain signal feature extraction is shown in Figure 4.

### 3.3. Frequency Domain Feature Extraction

For feature extraction in the frequency domain, we have generated a spectrogram by performing a short-time Fourier transform on the input signal, so the input for feature extraction is a two-dimensional spectral image. In recent years, deep learning has achieved great success in the field of image recognition. In order to present comprehensive information about the raw signal, improve the recognition rate and reduce the error rate, the spectral image of the fiber optic vibration signal is feature extracted by a two-dimensional convolutional neural network. Compared with other feature extraction methods such as shallow neural networks and artificial feature design, 2D-CNNs require fewer parameters to be considered and can automatically suppress the noise in the signal and extract the features related to vibration events more comprehensively. In order to meet the input of 2D-CNNs and reduce the computational effort, we rendered the STFT calculation results by using intensity interpolation light and dark processing and then downsampling, which reduces the feature loss caused by the downsampling process. Similar to the network structure in the time domain, we also use the ResNet structure for feature extraction. For the 2D-CNN, the size of the convolutional kernel affects the sensitivity of the results to different features. Larger convolutional kernels can improve the sensitivity of the model to global features, while smaller convolutional kernels are more pronounced for local features. After analyzing the vibration event spectrogram, we choose a convolution kernel size of 3 for the calculation.

### 3.4. Feature Fusion and Classification

A large number of studies have been conducted on signal feature extraction and recognition from the time domain and frequency domain, respectively. Their accuracy indicates that both time domain and frequency domain features have a certain validity, but also have certain shortcomings; therefore, the accuracy of their models cannot be further improved. Therefore, in this paper, we fuse time-domain features and frequency-domain features to achieve higher recognition accuracy. However, due to the different properties of time domain features and frequency domain features, the scale of the corresponding CNN-extracted features is also different, and similarly, the impact on the final classification results is also variable. In addition, there is a problem of data misalignment with two modes characterized throughout the method. For data containing multiple modes, such as time and frequency domains, one of the biggest challenges is to aggregate information from multiple modes to filter out the redundant parts of the modes while taking into account the complementary information. The Transformer structure has the following advantages: first, the Transformer only needs to flatten the features of the two modes, and then pass them into the Transformer for concat operation, which greatly simplifies the training. This greatly simplifies the difficulty of training. Second, the self-attention mechanism in the Transformer can ensure that the vector of each word of the output perceives the global information. Additionally, it solves the problem of long-distance dependent learning, which is of great importance in the task of signal classification. Third, the complexity of the Transformer is not high, which is very beneficial for the real-time performance of the end-to-end model. The Transformer structure is shown in Figure 5.

The multihead attention structure in Figure 5 is a combination of multiple self-attention structures. The formula for calculating a single self-attention is shown in Equation (Equation 5), where the input is the dimensionality of the feature vector extracted by the convolutional network and the query vector (Q), key vector (K), and value vector (V) obtained by multiplying the feature vector by three different weight matrices Wq, Wk, and Wv. The weights of the feature are calculated by Q and K. Finally, the V vector is weighted according to the calculated weights. In addition, the weights of each feature vector are normalized in order to ensure the stability of the gradient, which is achieved by dividing by $d_{k}$ and calculating the softmax activation. The multiheaded attention mechanism further refines the self-attention mechanism by stitching together different attentions and then performing a linear transformation, as shown in Figure 6.
(5)$$\begin{array}{c} Attention\left(Q,K,V\right)=Softmax\left(\frac{QK^{T}}{\sqrt{d_{k}}}\right)V \end{array}$$

The Transformer structure helps the whole model to understand which information in the time-frequency features of the vibration signal should be emphasized more in order to better integrate these features semantically. The final output feature *f* is calculated by Equation (Equation 6), where $\left\{w_{t}^{i}\right\}$ and $\left\{w_{f}^{i}\right\}$ are learnt by the Transformer.
(6)$$\begin{array}{c} f=\left\{w_{t}^{i}\right\}{\left\{c_{t}^{i}\right\}}_{i=1,2,\cdots ,n}+\left\{w_{f}^{i}\right\}{\left\{c_{f}^{i}\right\}}_{i=1,2,\cdots ,n} \end{array}$$

Finally, we use a fully connected layer with softmax to convert the output *f* into six classification probability classes.

## 4. Experimental Results and Evaluation

### 4.1. Network Training and Testing

In this article, our approach is implemented in Python 3.7 and TensorFlow 1.12 and runs on Windows 10 with an R7-4800H CPU. The neural network is trained with the following settings: the loss function uses the cross-entropy loss function, the optimization function uses the Adam algorithm, the batch size is 16, the learning rate is 0.01, and the epoch is 300 times.

In order to explore the optimal parameters of the model, we conducted several experiments, and the parameters explored include the input size for time domain feature extraction, the convolutional kernel size for the three channels of time domain feature extraction, and the depth of the ResNet network structure. The convolutional kernel size and input size were also explored in the network for frequency domain feature extraction, and since the spectrogram is generated based on the input size in the time domain, we changed the input size by downsampling the spectrogram instead of modifying the raw data size in order to ensure consistency with the time domain features. The details of the multimodal fusion recognition network are shown in Table 2. The parameters of Conv1d and Conv2d in feature extraction are the number of input channels, the number of output channels, and the size of the convolution kernel. The parameters of the Bottleneck are the number of input channels, the parameters of LSTM are the depth of LSTM, and the parameters of the TransformerEncoder are the depth and the number of heads of multiheaded self-attention. The parameters of the fully connected (FC) layer are the size of the input and the size of the output.

The learning curve is shown in Figure 7, which shows how the accuracy and loss values of the model change during the training process. After about 250 iterations, the fused multimodal fusion network starts to gradually converge and the accuracy gradually stabilizes, with the training accuracy reaching close to 100% and the test accuracy reaching 98.54%.

### 4.2. Results and Discussion

To demonstrate the performance of our proposed method, we performed the following ablation experiments and compared several previous research methods.
The time-domain features are extracted using a multiscale 1D-CNN, and the features extracted from CNNs of different scales are fused by the Transformer and finally classified by a fully connected layer;The frequency domain features are extracted using 2D-CNN, and the features are fused using the Transformer, and finally, the fully connected layer is used for classification;The features are extracted using a multiscale 1D-CNN and 2D-CNN for time domain and frequency domain data, respectively, and finally the features are fused and classified using a multilayer perceptron.

The results of the ablation experiments are shown in Table 3. In addition, we reproduced several previous studies for comparison. Since some of the effective features are inevitably lost when feature extraction is performed from the time domain or from the frequency domain alone, none of the accuracy rates exceed 96%, while in the third ablation experiment, we replaced the Transformer with a multilayer perceptron (MLP), and the accuracy rate decreased somewhat. This is because the number of operations required to correlate between two features of the Transformer does not grow with distance compared to the MLP, so it can better extract the semantic relationships between the features instead of directly combining the features by statistical means.

The key to artificial feature extraction-based methods is the combination of features selected empirically and targeted, which are usually obtained after rigorous mechanistic analysis of a class of signals, and thus can achieve high accuracy for specific domains, but the modeling time grows with the complexity of the recognition target and is less adaptable to different working conditions. Therefore the algorithm performs poorly on our dataset, with an accuracy of 84.22%. The increased depth of the traditional CNN makes the parameter optimization around the input layer slower and the pooling layer loses much valuable information, so even with parameter tuning and migration learning, the recognition rate is only 93.54% on our dataset. The AWT+2D-CNN structure is similar to the idea of ablation experiment 2, both use pre-processing to convert 1D signals into 2D images and use a CNN for learning, but the CNN is unable to combine features semantically, so the accuracy is slightly lower than our ablation experiment 2 at 94.78%. Compared with these studies, our proposed recognition algorithm with multimodal fusion can better extract temporal frequency features and perform semantic fusion, which can effectively improve generalization ability and accuracy, resulting in an increased recognition accuracy of 98.54%.

## 5. Conclusions

In this paper, an optical fiber vibration signal identification method based on multimodal feature extraction and fusion is proposed. The main contribution of this study is to explore the feature extraction method of vibration signals in both the time and frequency domains and to semantically fuse the features extracted in both the time and frequency domains, which are finally integrated into the monitoring system of long-distance pipelines. Compared with the traditional methods of convolutional neural networks and manual feature extraction, this model achieves better performance by fusing multimodal feature information through an attention mechanism. Thanks to the above advantages, this method achieves an accuracy of 98.54% on complex real-world application scenarios.

## Figures and Tables

**Figure 1 sensors-22-08795-f001:**
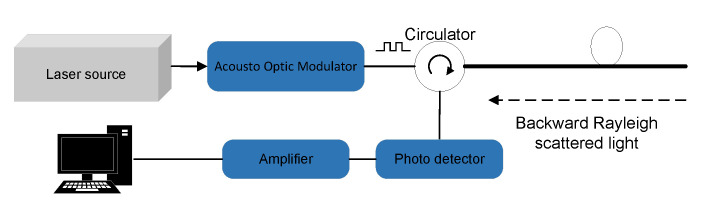
Φ-OTDR fiber optic system schematic.

**Figure 2 sensors-22-08795-f002:**
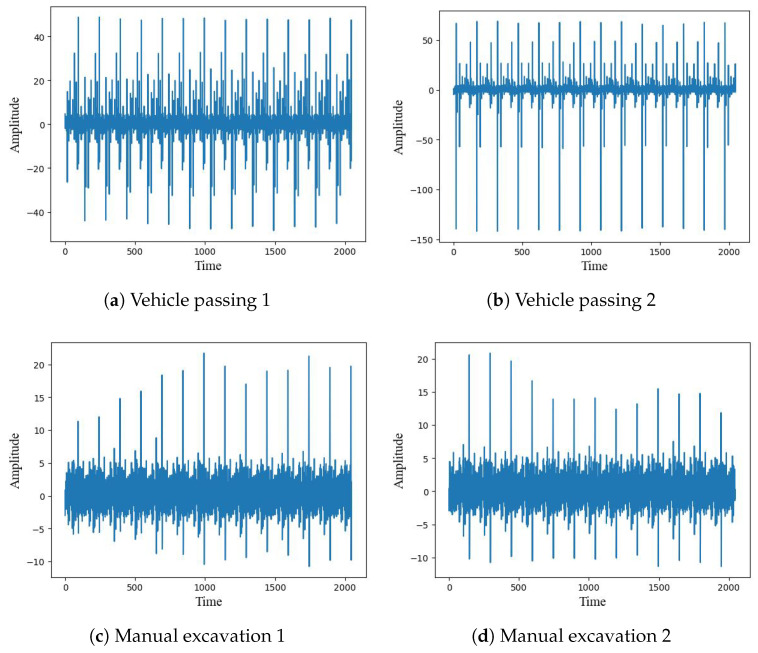
Comparison chart of different data segments. (**a**,**b**) are the data collected at the same location with different soil moisture conditions; (**c**,**d**) are the data collected at the same time with different locations.

**Figure 3 sensors-22-08795-f003:**
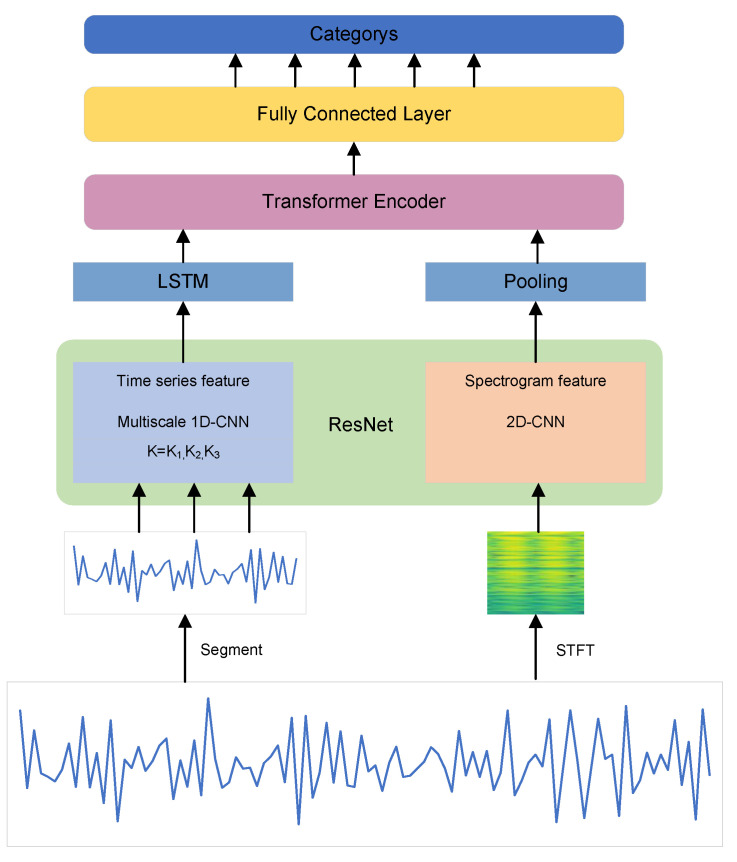
Network architecture of the proposed method.

**Figure 4 sensors-22-08795-f004:**
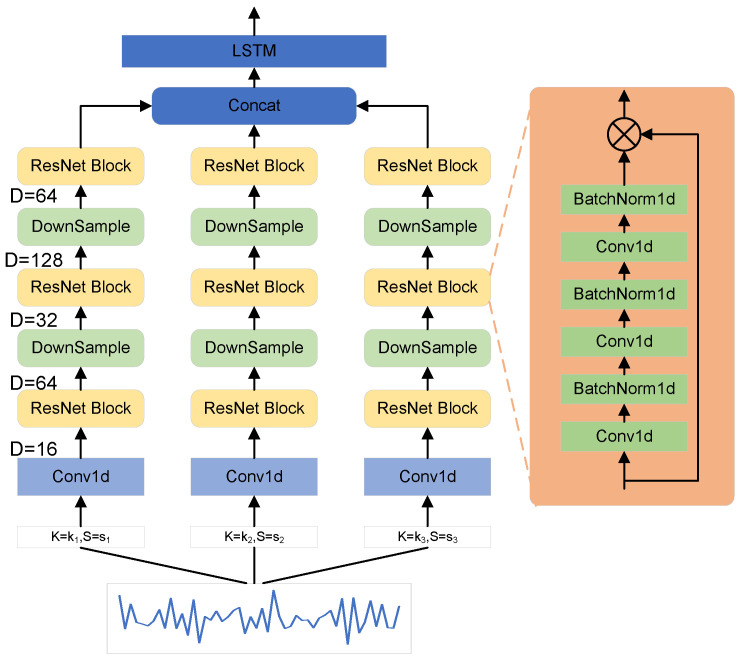
Time-domain feature extraction module. The nine residual blocks have the same composition and all consist of BottleNeck structures, where the three convolution kernels are of sizes 1, 3, and 1. The D in the figure indicates the data dimension of the corresponding layer.

**Figure 5 sensors-22-08795-f005:**
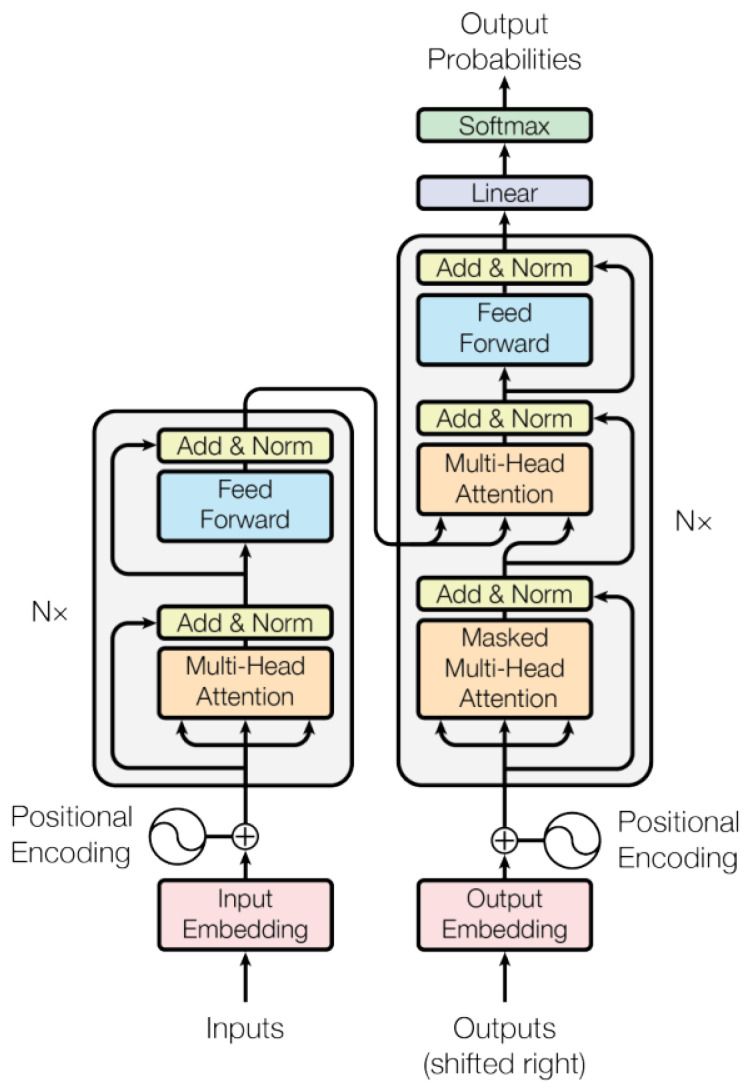
Transformer structure.

**Figure 6 sensors-22-08795-f006:**
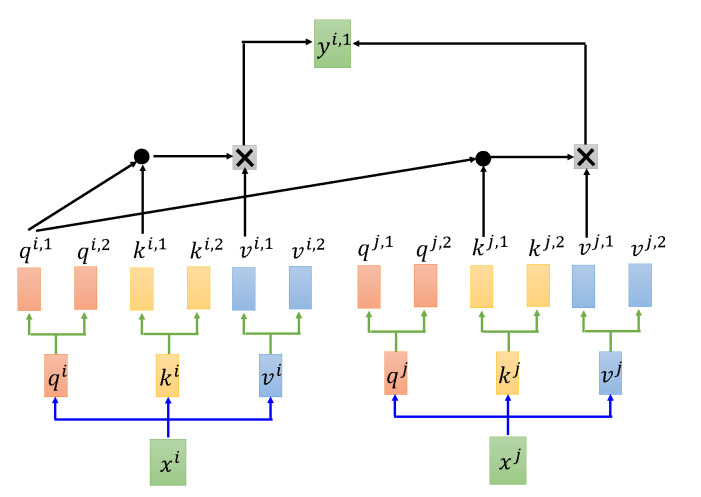
Multi-head attention.

**Figure 7 sensors-22-08795-f007:**
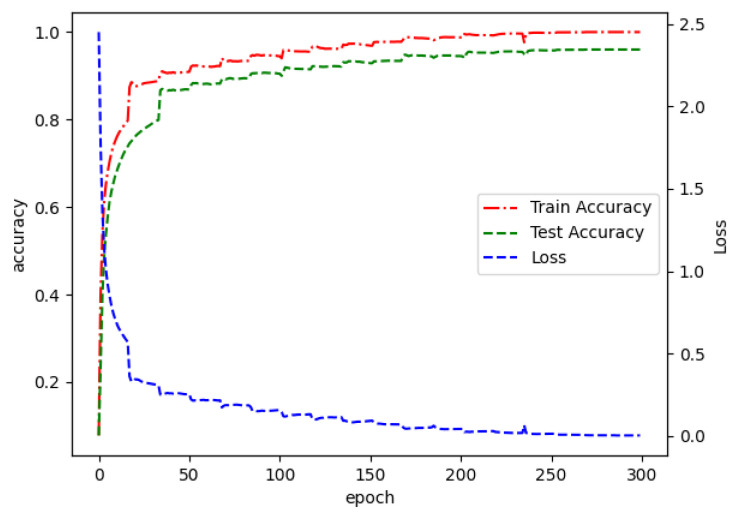
Learning curve.

**Table 1 sensors-22-08795-t001:** System parameters.

Item	Specification
Model	AQ1210A
Wavelength (nm)	1310 ± 25
Measuring distance range (km)	0.1∼256
Event blindness (m)	≤0.8
Pulse width (ns)	5∼20,000
Sampling resolution	Minimum 5 cm
Number of sample points	Up to 256,000
Loss measurement accuracy	±0.03 dB/dB

**Table 2 sensors-22-08795-t002:** Details of the multimodal fusion recognition network.

	Time Domain	Frequency Domain
Input	1×2048	3×90×90
Layer 1	Conv1d(1,16,[3,5,7])	Conv2d(3,16,3)
Layer 2	BatchNorm1d	BatchNorm2d
Layer 3	ReLU	ReLU
Layer 4	Bottleneck_1d(16) ×3 channel	MaxPool2d
Layer 5		Bottleneck_2d(16)
Layer 6		
Layer 7		
Layer 8		
Layer 9		
Layer 10	Bottleneck_1d(32) ×3 channel	
Layer 11		Bottleneck_2d(32)
Layer 12		
Layer 13		
Layer 14		
Layer 15		
Layer 16	Bottleneck_1d(64) ×3 channel	
Layer 17		Bottleneck_2d(64)
Layer 18		
Layer 19		
Layer 20		
Layer 21		
Layer 22	LSTM(3)	
Layer 23	Transformer Encoder(3,8)	
Layer 24	FC(64,32)	
Layer 25	FC(32,5)	

**Table 3 sensors-22-08795-t003:** Performance comparison of different methods.

Method	Description	Recognition Rate%
1D-CNN+Transformer	Ablation experiments with time-domain feature extraction	94.18%
2D-CNN+Transformer	Ablation experiments with frequency-domain feature extraction	95.2%
Time-frequency characteristics + MLP	Ablation experiments with MLP feature fusion	96.78%
TL1DCNN [14]	Ensembled transfer learning deep 1DCNN for feature extraction and recognition	93.54%
AWT+2D-CNN [18]	Acquisition of grayscale vibration images by wavelet transform and recognition by 2D-CNN	94.78%
SVM [11]	Recognition method combining SVM and artificial features	84.22%
**Proposed method**	Multimodal fusion based on attention mechanism	**98.54%**

## Data Availability

The data presented in this study are available on request from the corresponding author.

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
