# Peer review of "Φ-OTDR Signal Identification Method Based on Multimodal Fusion"

_sensors, 2022, doi:10.3390/s22228795_

Round 1

Reviewer 1 Report

The authors present their work on Φ-OTDR signal identification method based on multimodal Fusion. However, the focus of the paper is out of this journal’s scope.  The contents are purely related to neural network and artificial intelligence. The exception is Figure 2 which gives the captured sensor responses. But there is no explanation if these are phase response or amplitude response. Neither is Figure 2 quoted in the context. As the authors claimed that the main contribution of this study is to explore the feature extraction method of vibration signals in both time and frequency domains, and to semantically fuse the features extracted in both time and frequency domains. This could apply to many scenarios. From the method, experimental results, discussion, and conclusion I could hardly see how it improves this specific sensor performance or even be relevant to this sensor. 

 All the abbreviations should be defined or explained when first appearing such as SVM, ECG, 1D-CNN, LSTM, ResNet, etc. The paper is well written. There are several typos and mistakes. For example, in line 100 “The first term in Eq.2” should be “The first term on the right of Eq.2”; line 103: i th → ith? Line 118: sampling rate of 2k → sampling rate of 2kHz? line 216: resnet → ResNet? Lines 274-277: “The main contribution…. pipelines.” are duplicated.

Author Response

The main purpose of the work in this paper is to improve the correct rate of classification of the data collected by OTDR, avoiding the loss of features caused by operations such as filtering of the original signal. The data in Figure 2 is the original signal containing noise obtained after demodulation, and is also the input to the algorithm proposed in the later paper, which is the product of the analysis of traditional methods of recognition of OTDR signals combined with neural networks. In addition, since the OTDR system contains an FPGA computing unit, the method can be integrated directly inside the sensor system without the need for additional computing equipment.
The abbreviations and some typos and errors appearing in the paper have been corrected in the updated manuscrip.

Reviewer 2 Report

In this paper, the authors propose an end-to-end high-accuracy fiber optic vibration signal detection and identification algorithm by extracting features from the time domain and frequency domain by one-dimensional convolutional neural network and two-dimensional convolutional neural network, respectively, and introducing a self-attentive mechanism to fuse the features of multiple modes. This is rather interesting paper but some revisions should be provided:

1. By mentioning 'frequency domain' the authors refer the one not only to the term they meant but also to the OFDR system. Is it possible to use another term (just as a proposal, not a requirement)?

2. The CNN abbreviation is not explained in text after its first mentioning.

3. The paper operates with 'Ф-(phase)-OTDR' term, but the Figure 1 has no typical elements (like hybrid detector or something like this) to extract the phase of the signal. If the phase was extracted some another way, it should be described.

4. The experiments should be repeatable, so the elements of the setup need to be detailed in a separate table (model, manufacturer, etc)

5. Fig. 2 has the unsigned axes.

6. One more time, as a proposal. Is this possible to give also the methods operation time in the Table 2?

7. Please note that some of the paper sections ends with the drawings, I propose to place them above, after the first mentioning, within the text.

Round 2

Reviewer 1 Report

Figure 2 is not quoted in the context. Some abbreviations used are not consistent, for example: Fig1, Fig.6…, Eq1, Eq.2…

Author Response

There was a mistake where Figure 2 should have been cited, and this has now been fixed. All abbreviations have also been aligned. Please see the updated manuscript.